# Lack of Neuroprotective Effects of High-Density Lipoprotein Therapy in Stroke under Acute Hyperglycemic Conditions

**DOI:** 10.3390/molecules26216365

**Published:** 2021-10-21

**Authors:** David Couret, Cynthia Planesse, Jessica Patche, Nicolas Diotel, Brice Nativel, Steeve Bourane, Olivier Meilhac

**Affiliations:** 1UMR 1188 Diabète Athérothrombose Thérapies Réunion Océan Indien (DéTROI), Université de la Réunion, Inserm, Plateforme CYROI, F-97490 Sainte-Clotilde, France; cynthia.planesse@univ-reunion.fr (C.P.); jessica.patche@univ-reunion.fr (J.P.); nicolas.diotel@univ-reunion.fr (N.D.); brice.nativel@univ-reunion.fr (B.N.); steeve.bourane@inserm.fr (S.B.); olivier.meilhac@univ-reunion.fr (O.M.); 2Service de Neuroréanimation, Centre Hospitalo-Universitaire de La Réunion, 97410 Saint-Pierre de La Réunion, France; 3CIC-EC 1410, Centre Hospitalo-Universitaire de La Réunion, 97410 Saint-Pierre de La Réunion, France

**Keywords:** high-density lipoprotein, stroke, acute hyperglycemia, blood–brain barrier, hemorrhagic transformation

## Abstract

Introduction: The pleiotropic protective effects of high-density lipoproteins (HDLs) on cerebral ischemia have never been tested under acute hyperglycemic conditions. The aim of this study is to evaluate the potential neuroprotective effect of HDL intracarotid injection in a mouse model of middle cerebral artery occlusion (MCAO) under hyperglycemic conditions. Methods: Forty-two mice were randomized to receive either an intracarotid injection of HDLs or saline. Acute hyperglycemia was induced by an intraperitoneal injection of glucose (2.2 g/kg) 20 min before MCAO. Infarct size (2,3,5-triphenyltetrazolium chloride (TTC)-staining), blood–brain barrier leakage (IgG infiltration), and hemorrhagic changes (hemoglobin assay by ELISA and hemorrhagic transformation score) were analyzed 24 h post-stroke. Brain tissue inflammation (IL-6 by ELISA, neutrophil infiltration and myeloperoxidase by immunohisto-fluorescence) and apoptosis (caspase 3 activation) were also assessed. Results: Intraperitoneal D-glucose injection allowed HDL- and saline-treated groups to reach a blood glucose level of 300 mg/dl in the acute phase of cerebral ischemia. HDL injection did not significantly reduce mortality (19% versus 29% in the saline-injected group) or cerebral infarct size (*p* = 0.25). Hemorrhagic transformations and inflammation parameters were not different between the two groups. In addition, HDL did not inhibit apoptosis under acute hyperglycemic conditions. **Conclusion:** We observed a nonsignificant decrease in cerebral infarct size in the HDL group. The deleterious consequences of reperfusion such as hemorrhagic transformation or inflammation were not improved by HDL infusion. In acute hyperglycemia, HDLs are not potent enough to counteract the adverse effects of hyperglycemia. The addition of antioxidants to therapeutic HDLs could improve their neuroprotective capacity.

## 1. Introduction

The increasing incidence of obesity, diabetes, metabolic disorders, and associated risk factors worldwide is driving the development of cardiovascular diseases with clinical complications such as acute ischemic stroke (AIS) [1,2,3]. AIS remains a major cause of death and disability in our societies [4,5]. Intravenous thrombolysis by recombinant tissue plasminogen activator (rt-PA) is the gold standard of AIS management. Initially set at 3 h after the onset of symptoms, the therapeutic time window was extended to 4.5 h to allow more patients to be treated [6]. Mechanical thrombectomy can also treat a number of patients with proximal thrombosis [7,8]. However, this treatment is still limited to expert centers and to rather limited indications (only large vessels). Less than 5% of stroke patients are eligible to mechanical thrombectomy. It is important to continue the search for new neuroprotective treatments to enable more patients to be treated.

Hyperglycemia (HG), which is associated with worsening clinical outcomes [9], including greater infarct growth [10,11] and hemorrhagic transformation (HT) [12], is present in about 40% of patients with AIS [13]. Multiple mechanisms have been suggested in experimental studies, such as endothelial dysfunction, increased oxidative stress, and impaired fibrinolysis [14]. The American acute stroke guidelines suggest treating hyperglycemia to obtain a blood glucose level of 140 to 180 mg/dL (7.8–10.0 mmol/L) [15]. However, a recent large multicenter randomized trial (1151 patients included) failed to improve neurologic outcomes of stroke patient in spite of an intensive glucose control [16]. One of the reasons given by the authors for this failure was the frequency of severe hypoglycemia in the intensive treatment group. Furthermore, the multiple mechanisms by which HG may exacerbate brain damage, including endothelial dysfunction or increased oxidative stress, could have a persistent detrimental effect on the ischemic brain despite glycemic control at patient admission and should represent therapeutic targets for novel neuroprotective agents.

Due to their antioxidant, anti-inflammatory, anti-apoptotic, and anti-thrombotic properties, high-density lipoproteins (HDLs) represent a major anti-atherogenic factor beyond their reverse cholesterol transport effect, as they prevent atheroma formation and stabilizes plaques, preventing rupture and thrombosis [17]. HDL particles have a very complex protein and lipid structure, and the HDL-cholesterol (HDL-C) plasma concentration does not necessarily correlate to these protective effects, leading to the concept of HDL dysfunction [18]. We have shown this dysfunction in AIS condition in clinical setting [19]. More recently, we have demonstrated the neuroprotective effects of HDL therapy in a mouse stroke model by preserving the blood–brain barrier (BBB) integrity via the endothelial SR-BI [20]. The aim of the present study was to test HDL therapy in a model of cerebral ischemia associated with acute HG.

## 2. Results

### 2.1. Acute Hyperglycemia

Forty-two mice were subjected to a D-glucose (2.2 g/kg of body weight) intraperitoneal injection. This injection led to a similar increase in blood glucose between HDL- and saline-injected groups with a maximum value at the reperfusion time (Figure 1A, Saline: 306.30 ± 90.38 vs. HDL: 320.05 ± 81.92 mg/dL). During 90 min of brain ischemia, blood glucose levels ranged between 200 mg/dL and 300 mg/dL (Figure 1A red box). Twenty-two hours after middle cerebral artery occlusion (MCAO), blood glucose levels returned to baseline. The body weight was not different between groups (Figure 1B).

### 2.2. Mortality, Infarct Size, Hemorrhagic Transformation and BBB Leakage

In order to investigate the potential neuroprotective effect of HDL infusion during stroke under acute hyperglycemic conditions, 42 mice were subjected to 90 min of cerebral ischemia using the MCAO model. The mortality rate was lower in the HDL group than in the saline-infused group, although statistical significance was not reached (19% (4/21) versus 29% (6/21), respectively, *p* = 0.71). Of the 32 surviving mice, we chose to randomize 12 mice to be used for immunofluorescence analysis (six per group). The remaining 20 mice were randomized into two groups: 11 for the HDL group and 9 for the saline group. The infarct size evaluated by 2,3,5-triphenyltetrazolium chloride (TTC) staining and HT by macroscopic observations were subsequently quantified 24 h after ischemic stroke. We obtained a good interobserver correlation on infarct size and HT score (Supplemental Appendix A). The ischemic volume was not significantly different between saline- and HDL-injected mice (Figure 2A, Saline: 50.08 (IQR 45.11–56.11%), n = 9 vs. HDL: 49.62 (IQR 42.74–52.91%) n = 11; *p* = 0.25). Immunoglobulin G (IgG) infiltration was also analyzed as a marker of BBB permeability. IgG infiltration evaluated by the percentage of ipsilateral brain area labeled by IgG immunostaining was not different between the two groups (Figure 2B, Saline: 57.81 (IQR 49.6–63.2%) vs. HDL: 53.29 (IQR 46.8–55.7%) *p* = 0.13 n = 6 per group). In order to evaluate HT, we used an HT score (Figure 3A). This score was obtained by adding the scores from 0 to 4 on the five brain slices analyzed, which gives a score ranging from 0 to 20. Our results show that this score was not statistically different between saline- and HDL-treated mice (Figure 3B, Saline: 7.5 (IQR 5–12) n = 9 vs. HDL: 10 (IQR 9–11.5) n = 11; *p* = 0.28). Then, we analyzed hemorrhagic complications by quantifying hemoglobin content in the brain parenchyma. Firstly, we quantified hemoglobin infiltration in two brain regions (cortex and striatum) by immunofluorescence. Our results show that the percentage of hemoglobin staining in each brain areas analyzed were not significantly different between the two groups in both the cortex (Figure 3C, Saline: 0.52 ± 0.26% vs. HDL: 0.67 ± 0.61% of Hb staining/total of analyzed brain area; *p*:0.69 n = 6 per group) and in the striatum (Saline: 0.81 ± 0.47% vs. HDL: 0.53 ± 0.29% of Hb staining/total of analyzed brain; *p* = 0.47 n = 6 per group). Secondly, we quantified the total ipsilateral brain hemoglobin content using ELISA assay. This analysis confirmed that HT was not statistically different between saline- and HDL-treated mice (Figure 3D, Saline: 41.16 ± 12.55 mg/g vs. HDL: 49.5 ± 11.15 mg/g; *p* = 0.13). Taken together, these results showed that, in acute HG condition, HDL infusion failed to protect brain against ischemic damage including BBB leakage.

### 2.3. Neurological Evolution

At 24 h post stroke onset, neurological outcome was assessed using a 5 point Bederson’s scale [21] described in the Materials and Methods section. Neurological scores were not significantly different between the two groups (Saline: 2.3 (IQR 1.1–2.8) n = 9 vs. HDL: 2.5 (IQR 2–3) n = 11; *p* = 0.38).

### 2.4. Polymorphonulcear Cell Infiltration, MPO and IL-6 Quantification

To investigate peripheral immune cell infiltration and neuroinflammation, we quantified MPO-positive cells and IL-6 concentration in the ischemic brain. Our quantification of IL-6 concentration in brain extracts did not show any significant differences between the two groups (Figure 4A, saline: 28.55 ± 14.22 ng/g n = 9 vs. HDL: 40.68 ± 47.74 ng/g n = 11; *p* = 0.47). PMN infiltration was analyzed in two brain zones (cortex and striatum) (Figure 4B). The number of PMN positive cells was not significantly different between the two groups in both the cortex (saline: 41.58 ± 11.84 vs. HDL: 64.29 ± 29.99; *p* = 0.49 n = 6 per group) and in the striatum (saline: 13.18 ± 3.9 vs. HDL: 30.77 ± 9.77; *p* = 0.12 n = 6 per group). As observed for PMN infiltration, quantification of MPO concentration in ipsilateral brain was not significantly different between the two conditions (Figure 4, Saline 19.8 ± 4.6 µg/g n = 9 vs. HDL: 20.5 ± 5.3 µg/g n = 11; *p* = 0.7). These results indicate that HDL infusion did not protect the brain from PMN infiltration and inflammation during acute ischemia in HG condition.

### 2.5. Apoptosis Activity

As HDLs have been described to have an anti-apoptotic property, we quantified the number of cleaved caspase 3-positive cells in the ipsilateral brain area. Our results showed no statistical difference between the two groups (Figure 5A,B; Saline: 19.5 (IQR 9.7–51.2) vs. HDL: 44 (IQR 29.7–54.7); *p* = 0.13 n = 6 per group). This result suggests that, in our experimental conditions of acute hyperglycemia during stroke, HDLs did not exhibit anti-apoptotic effects.

## 3. Discussion

In our experimental stroke under acute HG conditions, we show that HDL infusion failed to protect the brain from ischemic lesions. These results are not in continuity with our previous study, in which we demonstrated that, under normoglycemic conditions, HDL infusion at the time of ischemia led to a reduction in infarct volume but also in hemorrhagic complications [20,22,23]. In the same mouse model of MCAO, we previously demonstrated that acute HG (without a history of diabetes mellitus) was sufficient to cause a poor neurological outcome and more brain damage [24]. In this first study, we highlighted that acute HG aggravated infarct volume and HT as early as 30 min after ischemia. These complications were correlated with the duration of ischemia and blood glucose levels [24]. Acute HG is known to be an independent factor of bad outcome after stroke. This acute elevation of blood glucose is related to a response to stress and a possible non-fasting state [9]. HG may have an antifribrinolytic effect and could lead to delayed reperfusion of the ischemic zone [25]. This detrimental effect of acute HG has also been demonstrated in patients who have undergone mechanical thrombectomy [26]. Many preclinical studies used HG models with blood glucose levels above 400 mg/dL [27]. These thresholds of HG represent only 0.5% of patients with acute ischemic stroke, and these patients are excluded from therapeutic strategies [6]. Similar to our previous study, our HG protocol permits us to maintain plasma glucose levels between 200 and 350 mg/dL during acute ischemic phase [24]. Hemorrhagic complications are also increased in acute HG [28,29]. Several mechanisms have been suggested, such as an increase in inflammation, oxidative stress, and free radical production [30,31]. Regardless of the mechanism, the primary cause of HT is the disruption of the BBB [29]. We have recently demonstrated that stroke per se could lead to an increase in brain inflammation [32]. This pro-inflammatory state, characterized among other things by an increase in pro-inflammatory cytokines, can lead to the production of adhesion molecules by endothelial cells and could promote leukocyte infiltration. Indeed, it was shown that neutrophil adhesion to the BBB may increase during the HG state [33].

HDLs are complex particles composed of multiple proteins, such as apolipoprotein A-I (apoA-I), phospholipids, and cholesterol esters. In addition to their ability to transport cholesterol from peripheral tissues back to the liver, they have pleiotropic effects including anti-oxidant, anti-apoptotic, anti-inflammatory, and anti-thrombotic properties [34]. HDLs were shown to have a neuroprotective effect in various mouse models of stroke [22,23]. Recently, we have demonstrated their beneficial effect in a mouse model of MCAO (4 h ischemia/reperfusion in normoglycemic conditions [20]. In the present study, we obtained a non-significant trend towards a decrease in cerebral infarct size after HDL infusion. The absence of a control group in this study makes it impossible to conclude with certainty about the relationship between HDL inefficiency and hyperglycemic status. We assume that these HDLs do not appear to be sufficiently potent to counteract the adverse effect of acute hyperglycemia on brain damage. HDL infusion failed to limit HT, PMN recruitment, and BBB leakage in our experimental conditions. Diabetes has a deleterious impact on HDL function [35]. Some studies have shown a decrease in the antioxidant activity of HDL particles isolated from patients with insulin resistance or with a high risk of atherosclerosis. In 2013, Sampaio et al. also demonstrated a blunted HDL protective functionality against LDL oxidation in type 1 diabetes patients [36]. Moreover, plasma glucose may be correlated with oxidized HDL (oxHDL) [37]. HDLs may undergo qualitative modifications due to glucose exposure, and a direct association has been observed between plasma glucose levels and apoA-I glycation [38,39]. These changes could lead to a decrease in the antioxidant properties of HDLs during acute HG and support the concept of a supplementation with functional HDL particles at the reperfusion in ischemic stroke. Endothelial cells seem to be particularly affected by the increased production of free radicals in response to the acute hyperglycemic episode [40]. Intracellular reactive oxygen species (ROS) production in endothelial cells is increased by acute HG at least through the activation of nicotinamide adenine dinucleotide phosphate NAD(P)H oxidase. In patients with diabetes, endothelial cell dysfunction due to increased superoxide production appears to be related to NAD(P)H oxidases but also endothelial nitrite oxide (NO) synthase uncoupling [41]. We have already demonstrated that acute HG was also responsible for an increase in the production of free radicals in cerebral endothelial cells [42]. This oxidative stress may participate in endothelial dysfunction, causing endothelial cells apoptosis and increased BBB permeability, contributing to worsening stroke [43]. In our experimental conditions, HDLs were isolated from the plasma of healthy volunteers and did not provide sufficient protection to limit the deleterious effects of ischemic stroke. These effects could be counteracted by antioxidant substances contained in plants such as polyphenols from the biodiversity of the Reunion Island [44]. HDLs could serve as carriers for potential hydrophobic protective molecules, in particular antioxidants such as curcumin. HDLs, particularly through SR-B1 receptors, bind and are taken up by endothelial cells [20]. This property would make it possible to focus on cerebral endothelial cells as a therapeutic target. Therapeutic HDLs could thus be improved with antioxidant components to limit the deleterious effect of HG on endothelial cells during stroke [44].

Our study has several limitations. First, one of the limitations of this study is the absence of a normoglycemic group treated with the same batch of HDLs, which would attest to their protective effects in the absence of induced hyperglycemia. Despite a trend toward reduced infarct volume and IgG infiltration (Figure 2A,B) in the HDL-treated group, we cannot conclude with certainty whether this lack of protection is due to the hyperglycemic condition or to a lack of functionality of the HDLs tested. An additional group without hyperglycemia induction would have been appropriate but would have had to be performed in a different experimental setting (4 h instead of 1.5 h MCAO) to produce a cerebral infarct with of sufficient volume to test the effects of HDLs [24,44]. However, reproducibility of HDL isolation was tested in terms of composition and potential contamination by SDS-PAGE and Coomassie blue staining. In future studies, a normoglycemic control group and/or in vitro experiments attesting to the functionality of HDLs would be required. Second, the inflammation observed in the HDL group could be due to the injection of HDLs. Indeed, we cannot exclude a sterility problem of the injected solution. However, in our study, there is only a non-significant trend towards increased inflammation in the HDL group. Finally, we chose to test a single dose of HDL as therapeutic. We established this dose based on our previous work and studies conducted in the clinical setting.

In this study, we have shown that HDL infusion in a 90 min ischemia/reperfusion mouse model in HG condition failed to protect brain against reperfusion injury. Acute HG could overwhelm the neuroprotective effects of HDLs. Many stroke patients undergo acute hyperglycemia. HDL does not appear to be potent enough in this specific condition.

## 4. Materials and Methods

### 4.1. Acute Hyperglycemia

Intraperitoneal (IP) injection of 2.2 g of glucose/kg body weight was performed 20 min before MCAO, as previously shown [24]. Blood was sampled by tail puncture at different time points: before surgery and treatment (baseline), before MCAO, 1 h after MCAO, reperfusion time, 1 h after reperfusion, and before euthanasia to measure blood glucose levels (Hemoglucotest, Accu-Chek©, Roche Diabetes Care GmbH, Germany).

### 4.2. Isolation of High-Density Lipoproteins

Lipoproteins were isolated by ultracentrifugation from plasma of healthy volunteers. Plasma density was adjusted to d.1.22 with KBr (KBr: 2.8 mol/L or 334.5g/L) and then overlaid with KBr saline solution (d.1.063) (KBr: 2.96 mol/L or 352.2 g/L). Ultracentrifugation was performed at 100,000 g for 20 h at 10 °C. The density of the bottom fraction containing HDLs was adjusted to 1.25 with KBr (KBr: 0.79 mol/L or 93.6 g/L), and this solution was overlaid with KBr saline solution (d.1.21). A second step of ultracentrifugation at 100,000 g for 20 h at 10 °C was performed and the HDL fraction (yellow top layer) was recovered as a single band that was then extensively rinsed with PBS (5 times the initial volume) and concentrated using a centrifugal concentrating device (cutoff 10 kDa).

### 4.3. Mouse Model of Focal Cerebral Ischemia and Experimental Protocol

In vivo experiments were conducted in accordance with the French and European Community Guidelines for the Use of Animals in Research (86/609/EEC and 2010/63/EU) and approved by the Ethics Committee for animal experimentations (CYROI APAFlS#2643-2015100621507613 v2. Forty-two C57BL/6J male mice of 10 weeks old were purchased from Janvier Labs and maintained under a standard 12 h light/12 h dark cycle with free access to water and standard chow diet [45]. Mice were randomized and anesthetized with isoflurane and their body temperatures maintained on a heating pad throughout. The animals were infused intraperitoneally with 2.2 g of glucose per kg of body weight. Cerebral ischemia/reperfusion was induced by a 90 min intraluminal middle cerebral artery occlusion (MCAO) by introducing a 7–0 silicon-rubber-coated monofilament (702056PK5; Doccol Corporation, MA, USA) into the right common carotid at the bifurcation of the right MCA and the right internal carotid. The diameter of the monofilament (0.20 +/− 0.01 mm) allows the complete obstruction of the MCA of 25 g mice [24]. After removing the monofilament, reperfusion was allowed and randomly injected directly in the right common carotid (HDL group: 10 mg/Kg of ApoA-I or saline group: same volume of saline). Computer based randomization was used to allocate animals to each group. Experiments were blinded and the operator was unaware of the group during surgery and final analyses. Based on previous data from our group, showing that, under these experimental conditions, a statistical significance can be reached with 8 WT mice per group, and given a mean 30% mortality rate, the number of mice included was set at 12, yielding 80% statistical power to detect an absolute difference of 5 in HT scores for a 2-tailed alpha level of 0.05 using a t-test (with known mean HT score of 10 and 3.5 of SD) [24].

### 4.4. Infarct Size and Hemorrhagic Transformation Score Evaluation

#### 4.4.1. Infarct Volume Assessment

At 22 h after the reperfusion, mice were euthanized by intracardiac puncture, and intravascular washout was performed by intracardiac perfusion of saline under anesthetic to avoid pain during puncture. Animals were randomly allocated to TTC or immunostaining analysis groups. Brains were removed and cut into 5 serials of 1 mm coronal slices using a brain matrix mold for evaluation of infarct volumes and hemorrhagic transformations (HT) (hemorrhagic score and extravascular hemoglobin). The infarct volume was determined on coronal brain sections stained with a 2% solution of 2,3,5-triphenyltetrazolium chloride (TTC) for 20 min at room temperature. Infarct volume was assessed by two independent observers blinded to the group status by using Image-J^®^ (image-processing software), as previously described [32]. Briefly, infarct volume (IV) was calculated from the volume of the normal grey matter in the control contralateral hemisphere (*V**CH*) and in the lesioned ipsilateral hemisphere (*V**IH*), two parameters that are unaffected by the presence and extent of edema in the infarcted area, using the following formula *IV* (%) = (*V**CH* − *V**IH*)/*V**CH* × 100. We also use this calculation method to quantify IgG infiltration in coronal sections after immunofluorescence staining.

#### 4.4.2. Hemorrhagic Transformation Score Assessment

Hemorrhagic transformation (HT) was macroscopically scored on an arbitrary scale from 0 to 4 on coronal brain slices by 2 independent operators blinded to the animal treatment. This macroscopic HT score based on the definitions used in human clinical studies is also often used in preclinical models. Hemorrhages were classified into five types for each coronal brain sections: (0) no hemorrhage; (1) hemorrhagic infarction type 1 (small petechial hemorrhagic infarction HI-1); (2) hemorrhagic infarction type 2 (confluent petechial hemorrhagic infarction HI-2); (3) parenchymal hematoma type 1 (<30% of infarct, mild mass effect PH-1); and (4) parenchymal hematoma type 2 (>30% of infarct, marked mass effect PH-2) [46]. This score was obtained by adding the scores from 0 to 4 on the 5 brain slices analyzed, which gives a score ranging from 0 to 20 [22,46].

### 4.5. Neurological Evaluation and Mortality

Neurological outcome was blindly evaluated by two independent observers, 22 h after reperfusion, using a five-point Bederson’s scale: 0, no deficit; 1, mild forelimb weakness; 2, severe forelimb weakness, consistently turns to side of deficit when lifted by tail; 3, compulsory circling; 4, unconscious; and 5, death [21]. The mean of the two results was included in the final analysis.

In the mortality analysis, only deaths after the MCAO procedure were considered.

### 4.6. Immunofluorescence Assay

#### 4.6.1. Histology

For the histological analysis, mice were perfused with 4% paraformaldehyde (PFA) in PBS prior to dissecting the brain. The brains were postfixed overnight in 4% PFA at 4 °C and washed 3 times 10 min with cold PBS. The tissues were cryoprotected overnight at 4 °C in 30% sucrose-PBS. The brains were then embedded in OCT solution and cut into 14 μm sections on a cryostat and collected onto glass slides.

#### 4.6.2. Hemoglobin, Neutrophil Detection, IgG and Caspase-3 Detection

Bain sections were incubated overnight with primary antibodies at 4 °C. Primary antibody staining was detected with fluorophore-conjugated secondary antibodies. Images were captured using a NanoZoomer S60 Digital slide scanner (Hamamatsu, Japan). Cell counting (neutrophil and caspase3) and immunostaining area (hemoglobin and IgG) were performed by 2 independent operators blinded to the treatment of the animals by using ImageJ software with thresholds set according to signal intensity. For the evaluation of Hb staining in the infarcted brain, we quantified the percentage of area of Hb immunostaining by using ImageJ software (2 different zones were chosen (1: cortex and 2: striatum) with 5–7 sections analyzed per mice). The following primary antibodies were used: rabbit anti-hemoglobin (1:200, ab191183, Abcam, Paris, France), rat anti-neutrophil (1:200, ab2557, Abcam, Paris, France), Alexa-Fluor 594 Donkey anti-mouse IgG (1:500, X0931, Dako, Agilent Technologies, Santa Clara, CA, USA), and rabbit anti-caspase-3 (1:100, ab13847, Abcam, Paris, France). The following secondary antibodies were used: Alexa-Fluor 488 Donkey anti-Rabbit (1:1000, A-21206, invitrogen Molecular Probes, Invitrogen, Carlsbad, CA, USA) and Alexa-Fluor 594 Donkey anti-Rat (1:500, A-21209, invitrogen Molecular Probes, Invitrogen, Carlsbad, CA, USA).

### 4.7. Enzyme-Linked Immunosorbent Assay for Hemoglobin in the Brain

Hemoglobin (Hb), myeloperoxidase (MPO), and interleukin (IL-6) content was measured by ELISA (Mouse Hemoglobin ELISA Kit, ab157715—Mouse MPO ELISA kit, ab155458; Abcam, Paris, France—Mouse IL-6 ELISA Kit, 88-7064-88, eBioscience, Santa Clara, USA) using brain homogenates prepared from TTC sections. Hb, MPO, and IL-6 concentration values were reported on total protein content from the ischemic hemisphere.

### 4.8. Statistics

Statistical analyses were performed using GraphPad Prism 5 software (La Jolla, CA, USA). Results are expressed as mean ± standard error or median with interquartile range. Comparison between multiple groups were performed by one-way analysis of variance (ANOVA, Kruskall Wallis), followed by Mann–Whitney tests. A value of *p* < 0.05 was considered statistically significant. For in vivo experiments, data are presented as median with interquartile range. Data were analyzed by a Kruskal–Wallis test followed by a Dunn’s multiple comparison test if *p* < 0.05. A 2-tailed value of *p* < 0.05 was considered significant. Fisher’s exact test was used for mortality analysis. Pearson correlation coefficients were used to analyze interobserver correlation in the assessment of infract size and HT.

## Figures and Tables

**Figure 1 molecules-26-06365-f001:**
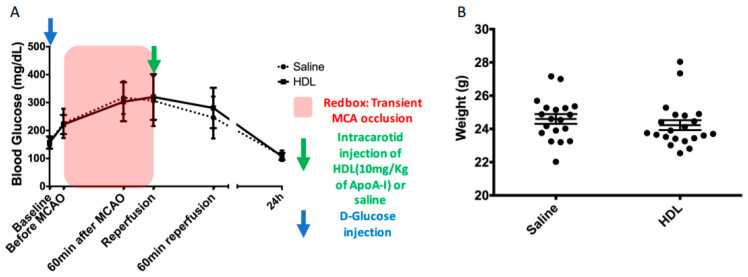
Timeline of acute hyperglycemia, MCAO, and intracarotid injection procedures. Body weight of mice before surgery. (**A**): IP injection of D-glucose at the initial time (blue arrow). Plasma glucose levels of saline- and HDL-injected mice during the procedures. The red box represents the ischemic period. The green arrow indicates the time of intracarotid injection of HDLs or saline. (**B**): Body weights of mice were not different between groups.

**Figure 2 molecules-26-06365-f002:**
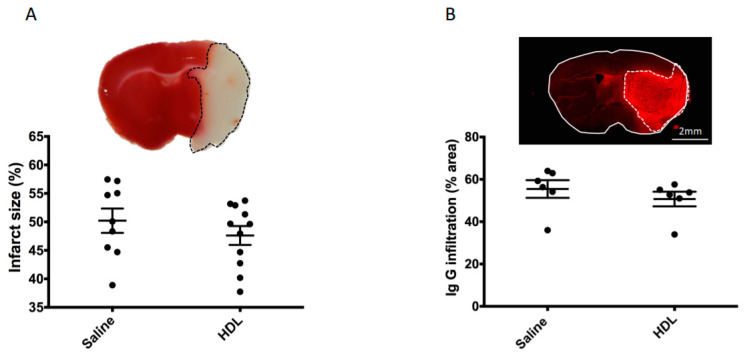
Infarct size and blood–brain barrier leakage. (**A**): Coronal brain sections stained with TTC. The red zone corresponds to the healthy brain region, whereas the white zone represents the infarcted area. No statistical difference was observed between saline- and HDL-treated mice (Saline: n = 9, HDL n = 11). (**B**): Coronal brain sections stained by with an antibody against immunoglobulins G (IgG) (red). The bright red area represents the infiltration of the brain parenchyma by IgG in the infarcted area. No statistical difference was observed between saline- and HDL-treated mice (Saline: n = 6, HDL n = 6).

**Figure 3 molecules-26-06365-f003:**
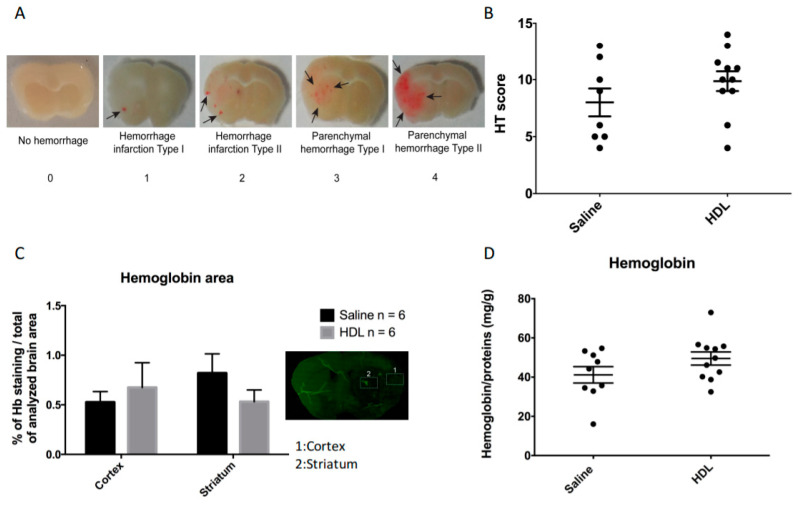
Hemorrhagic transformation (HT) and hemoglobin quantification. (**A**) Representative brain slice pictures allowing HT grading: no hemorrhage rated 0, hemorrhage infarction type I, characterized by small petechiae rated 1, hemorrhage infarction type II, characterized by confluent petechiae rated 2, parenchymal hemorrhage type I, characterized by an area <30% of infarct with mild mass effect rated 3, and parenchymal hemorrhage type II, characterized by an area >30% of infarct with marked mass effect, rated 4. (**B**) HT score obtained by adding the score of each of the five coronal sections analyzed per mice (0 to 20). Saline: n = 9, HDL: n = 11. (**C**) Percentage of hemoglobin immunostaining in two brain areas: a superficial area, the cortex (1), and a deep area, the striatum (2). Saline: n = 6, HDL: n = 6. (**D**) ELISA quantification of total hemoglobin on the infarcted brain region. Saline: n = 9, HDL: n = 11.

**Figure 4 molecules-26-06365-f004:**
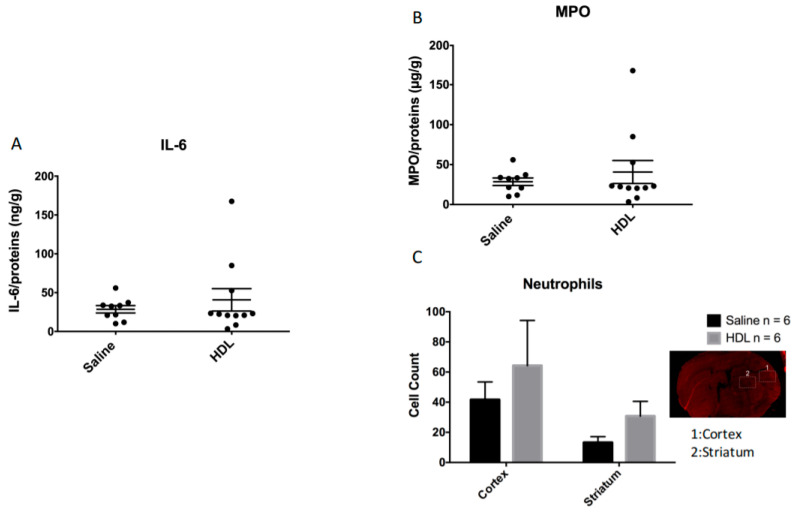
Brain inflammation in acute HG condition. (**A**): Interleukin-6 (IL-6) quantification by ELISA on the infarcted brain region. Saline: n = 9, HDL: n = 11. (**B**): Myeloperoxidase (MPO) quantification by ELISA on the infarcted brain region. Saline: n = 9, HDL: n = 11. (**C**): The number of neutrophil positive cells per selected field was evaluated in two areas of the ischemic brain: the cortex (1) and the striatum (2). Saline: n = 6, HDL: n = 6.

**Figure 5 molecules-26-06365-f005:**
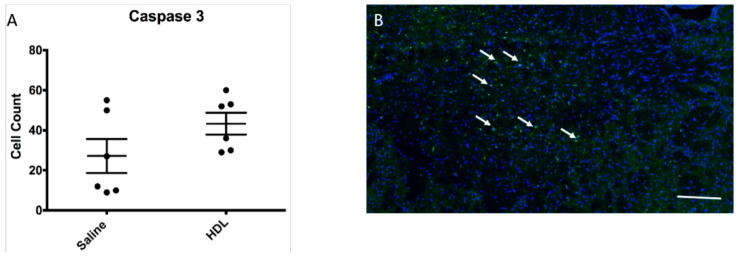
Apoptosis in infarct area. (**A**) Mean number of cleaved caspase 3 positive cells in selected field of infarcted brain. Saline: n = 6, HDL n = 6. (**B**) The white arrows show positive cleaved caspase 3 positive cells (green). Cell nuclei are stained with DAPI (blue). Bar = 80 µm for low magnification picture.

## Data Availability

The data presented in this study are available on request from the corresponding author.

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
