# Peer review of "Lack of Neuroprotective Effects of High-Density Lipoprotein Therapy in Stroke under Acute Hyperglycemic Conditions"

_molecules, 2021, doi:10.3390/molecules26216365_

Round 1

Reviewer 1 Report

The paper builds upon the author´s previous work investigating the potential use of HDL as therapy in middle cerebral artery occlusion (MCAO).

While the authors present novel findings that protective capabilities of isolated human HDL are not able to counter the effects of acute hyperglycemia, the paper lacks additional experimental input to support those claims.

Major concerns:

1.) The authors have used WT animals in their experiments, which have endogenously high levels of circulating HDL. Are the authors aware of local concentrations of native vs artifically injected HDL in these animals? What is the half-life of injected HDL in plasma and brain. The authors might consider using ApoA1 KO animals (lacking endogenous HDL). Exogenously added HDL could show stronger effects compared to the saline group in HDL-deficient mice.

2.) The authors themselves nicely explain, that the quantity of HDL does not necessarily correlate with its protective effects. Better (and pharmacologicaly more relevant) results could be obtained treating the animals with reconstituted HDL or even ApoA1 mimetics, which are in clinical trials.

3.) Some markers of inflammation, such as neutrophils infiltration are slightly (but not significantly) elevated following HDL injection. Is this a sterility issue? Would same trend be observed if murine HDL were to be isolated and injected into mouse brain.

Minor: 

1.) Figures need to be proof-checked. Order (left-right- top -bottom) of subfigures as well as Figure 1 labels. X-axis in figure one should show time (constant variable) in hours, with additional treatment options added to the appropriate time points at top/bottom of the figure. All abreviations in figures, should be explained in figure legends (e.g. Neutro). Figure text should be the same size (e.g. Fig. 3). 

Author Response

Reviewer 1

The paper builds upon the author´s previous work investigating the potential use of HDL as therapy in middle cerebral artery occlusion (MCAO).

While the authors present novel findings that protective capabilities of isolated human HDL are not able to counter the effects of acute hyperglycemia, the paper lacks additional experimental input to support those claims.

Major concerns:

1.) The authors have used WT animals in their experiments, which have endogenously high levels of circulating HDL. Are the authors aware of local concentrations of native vs artifically injected HDL in these animals? What is the half-life of injected HDL in plasma and brain. The authors might consider using ApoA1 KO animals (lacking endogenous HDL). Exogenously added HDL could show stronger effects compared to the saline group in HDL-deficient mice.

We agree with the Reviewer that mice have high levels of HDLs. It is difficult to quantify the local HDL concentration within the infarcted area. Considering that the concentration of circulating HDL-C in male C57/BL6 mice is approximately 65 mg/mL [1], an intra-arterial injection of 10 mg ApoA1/kg (0.25 mg/mouse) would only slightly increase HDL-C concentration to 71 mg/mL. Although this HDL-C supplementation appears low, this injection is performed directly into the carotid artery and ApoA1 is expected to reach directly the infarcted area. This concentration was sufficient to provide a beneficial effect in our previous studies [2]. The half-life of injected HDLs is approximately 6-24 hours but there is no data available for the brain, and following injection, HDL particles are rapidly taken-up by the liver (within a few hours). We have shown in this study that HDL particles could reach the stroke area and be taken up by brain cells. However, it is very difficult to determine the local concentration of HDLs. We agree with the Reviewer that ApoA1 KO mice would be appropriate to enhance the beneficial effects of an HDL injection. However, our previous studies to which we refer have shown protective effects of HDL injected by the arterial route. The lack of beneficial effects in the current study underscores the aggravating effect of hyperglycemia and provides us with a good model to improve HDL functionality via their enrichment with additional protective molecules (eg, antioxidants), as discussed.

2.) The authors themselves nicely explain, that the quantity of HDL does not necessarily correlate with its protective effects. Better (and pharmacologicaly more relevant) results could be obtained treating the animals with reconstituted HDL or even ApoA1 mimetics, which are in clinical trials.

We agree with the reviewer that it would be interesting to test reconstituted HDL or ApoA1 mimetics, as this could be more easily translated to the clinic. However, these products are not commercially available, in particular the rHDLs that are currently used in clinical trials. In addition, we hypothesize that the beneficial effects of HDLs do not rely only on apoA1, but also on other protective molecules such as sphingosine 1-phosphate or paraoxonase. Finally, in our previous benchmark study, we used HDLs isolated from plasma that showed protective effects [3]. The testing of rHDLs and ApoA1 mimetics could be the subject of a new study, which could represent a prerequisite for moving into the clinic in humans.

3.) Some markers of inflammation, such as neutrophils infiltration are slightly (but not significantly) elevated following HDL injection. Is this a sterility issue? Would same trend be observed if murine HDL were to be isolated and injected into mouse brain.

HDLs were isolated from plasma with precautions during the ultracentrifugation and dialysis process to avoid contamination. The preparation filtered with a 0.2-micrometer filter before injection.  However, a sterility issue cannot be totally excluded and this limitation has been added in the new version of the manuscript. Intravenous and potentially intra-arterial injection of human HDLs in mice can elicit an immune response but only after 10 days, precluding further injections after that time. Finally, it is technically almost impossible to obtain enough murine HDLs to perform such a study.

“Second, the inflammation observed in the HDL group could be due to the injection of HDLs. Indeed, we cannot exclude a sterility problem of the injected solution or an immune reaction. However, in our study there is only a non-significant trend towards increased inflammation in the HDL group”

Minor: 

  • Figures need to be proof-checked. Order (left-right- top -bottom) of subfigures as well as Figure 1 labels. X-axis in figure one should show time (constant variable) in hours, with additional treatment options added to the appropriate time points at top/bottom of the figure. All abreviations in figures, should be explained in figure legends (e.g. Neutro). Figure text should be the same size (e.g. Fig. 3). 

Thank you for this comment that improves the clarity of our trial. Figures have been modified in the revised version of the manuscript.

  1. Paigen, B.; Holmes, P.A.; Mitchell, D.; Albee, D. Comparison of atherosclerotic lesions and HDL-lipid levels in male, female, and testosterone-treated female mice from strains C57BL/6, BALB/c, and C3H. Atherosclerosis 1987, 64, 215-221, doi:10.1016/0021-9150(87)90249-8.
  2. Lapergue, B.; Moreno, J.A.; Dang, B.Q.; Coutard, M.; Delbosc, S.; Raphaeli, G.; Auge, N.; Klein, I.; Mazighi, M.; Michel, J.B., et al. Protective effect of high-density lipoprotein-based therapy in a model of embolic stroke. Stroke; a journal of cerebral circulation 2010, 41, 1536-1542, doi:10.1161/STROKEAHA.110.581512.
  3. Tran-Dinh, A.; Levoye, A.; Couret, D.; Galle-Treger, L.; Moreau, M.; Delbosc, S.; Hoteit, C.; Montravers, P.; Amarenco, P.; Huby, T., et al. High-Density Lipoprotein Therapy in Stroke: Evaluation of Endothelial SR-BI-Dependent Neuroprotective Effects. Int J Mol Sci 2020, 22, doi:10.3390/ijms22010106.

Reviewer 2 Report

Minor

  1. Lines 43-46 implies that IV thrombolysis is synonymous with mechanical thrombectomy. Please amend to ensure dichotomy between the two reperfusion techniques.
  2. Assessment of blood glucose levels is not reported in the methods (i.e.., techniques, volumes).
  3. Figure 1 injection of D-glucose needs to be indicated on graph. Reference to red box not present on figure. 
  4. Line 92, provide quantitative statistics in addition to %.
  5. It would be advantageous to report the splitting of experimental groups into TTC and immuno in the methods section. 
  6. Units of measurment is absent from a number of results. 
  7. It would be valuable to provide any information on the species differences between human and rodent HDL.
  8. Line 252: please provide molarity/concentration of KBr.
  9. Were all animals fasted prior to surgery?
  10. Was intracardiac euthanasia performed under anaesthetic?
  11. Consider providing interrater reliability of infarct measurments and HT scores as supplementary information. 
  12. given TTC brain slices should be electronically stored. It would be valuable to provide an alaysis of HT volume rather than a human specific score of 0-4. 
  13. Line 316 implies neurological scores are performed pre-surgery. 

Major

  1. The authors offer a model of hyperglycaemia intended to reflect hyperglycaemic conditions in clinical stroke patients. It is well accepted that hyperglycaemia in stroke patients is of a chronic nature (over extended periods of time) due to lifestyle factors. I question if an acute hyperglycaemic event presented by the Authors is representative of the additional physiological parameters assosciated with chronic hyperglycaemia in stroke patients. At a minimum, this should be discussed in the discussion.
  2. When reporting the statistics of haemorrhgaic transformation, how can a score of >4 be achieved if the HT score only goes to 4?
  3. When determining if HDL is neuroprotective, it would be advisable that a dose reponse be performed, especially when examining HDL's efficacy in a new model (hyperglycaemia + MCAO). Using only a single dose and stating HDL is not neuroprotective undervalues the potential efficacy HDLs could have if a higher dose was used. 
  4. Why was an intra-carotid injection used to administer HDLs over IV or IP? I have reservations that the bolus injection (versus slow infusion) of either HDLs or saline could contribute to haemorrhagic transformation post-reperfusion due to significant pressure differentials in a compromised vascular enviroment. This becomes an important point when demonstrating HT values are similar between treatment and vehicle. 
  5. I do not understand the value of examining IgG independent of infarct when there is an implied dependency of loss of BBB integrity within the region of the infarct by nature of cell death; this is supported by similar % area values for infarct and IgG presented in Fig 2). Ideally, IgG should be examined and correlated with infarct within the same animal. To perform this, I would recommend infarct evaluation via MRI followed by IgG staining. I understand this is outside the scope of the current study, however, the major comment is required to be addressed. 

Author Response

Reviewer 2

Minor

  1. Lines 43-46 implies that IV thrombolysis is synonymous with mechanical thrombectomy. Please amend to ensure dichotomy between the two reperfusion techniques.

Thank you for this comment. The correct sentence has been added to the revised version of the manuscript. “Less than 5% of stroke patients are eligible to mechanical thrombectomy”

  1. Assessment of blood glucose levels is not reported in the methods (i.e.., techniques, volumes).

We agree that this sentence was detailed enough. We added: “by tail puncture” and “Accu-Chek©”

  1. Figure 1 injection of D-glucose needs to be indicated on graph. Reference to red box not present on figure. 

Figure 1: Acute hyperglycemia procedure and timing of MCAO and intracarotid infusion

A: After IP injection of DGlucose at the initial time (blue arrow), plasma glucose levels increase to 300mg/dL at 60min after MCAO for both groups and return to baseline 24 hours later. The red box represents the ischemic period. The green arrow indicates the time of intracarotid infusion of HDL or Saline.

B: Body weight was the same for both groups.

  1. Line 92, provide quantitative statistics in addition to %.

Thank you. Statistics have been added to the revised version. “p=0.71” and “Fisher’s exact test was used for mortality analysis”

  1. It would be advantageous to report the splitting of experimental groups into TTC and immuno in the methods section. 

We totally agree with the Reviewer #2. This sentence has been added: “Animals were randomly allocated to TTC or immunostaining analysis groups”

  1. Units of measurment is absent from a number of results. 

Thank you for this comment. We added units of measurement when necessary in the revised version. 

  1. It would be valuable to provide any information on the species differences between human and rodent HDL.

There are no major differences in human vs mouse HDL particles in terms of protein and lipid contents. The major apolipoproteins are found in both HDLs [4]. Different studies report that the main effects of HDLs are mediated by apoA1, which is present in similar abundance in mouse and human HDL particles.

  1. Line 252: please provide molarity/concentration of KBr.

d=1.21 (KBr : 2.8 mol/L or 334.5g/L)

d=1.22 (KBr : 2.96 mol/L or 352.2g/L)

d=1.063 (KBr : 0.79 mol/L or 93.6g/L)

  1. Were all animals fasted prior to surgery?

Thank you for this comment. The animals were not fasted before surgery, in order to reproduce the real conditions of hyperglycemic stress during stroke.

  1. Was intracardiac euthanasia performed under anaesthetic?

Yes, absolutely, we chose to perform intracardiac euthanasia under anesthesia to avoid pain during puncture. We have modified the sentence in the materials and methods section to improve clarity.

  1. Consider providing interrater reliability of infarct measurments and HT scores as supplementary information.

Supplementary Figure S1: Inter Operator agreement

A : We have a good inter-operator correlation for the evaluation of the ischemic volume with R= 0.89

B : We also have a good inter-operator correlation for the evaluation of the HT score with R= 0.76

  1. given TTC brain slices should be electronically stored. It would be valuable to provide an alaysis of HT volume rather than a human specific score of 0-4. 

Thank you for the comment that helped us improve the clarity of our study. We chose to use 3 different methods to assess HT. We used macroscopic assessment with the HT score based on the ECASS clinical trial, quantification of hemoglobin area after immunofluorescence staining, and quantification of Hb with the ELISA method.

  1. Line 316 implies neurological scores are performed pre-surgery. 

We agree with the referee#2, this sentence is imprecise. We have added this sentence to the text: " In the mortality analysis, only deaths after the MCAO procedure were considered.” Neurological scores are performed 24h after surgery and just before euthanasia:” At 24h post stroke onset, neurological outcome was assessed using a 5-point Bederson’s scale”.

Major

  1. The authors offer a model of hyperglycaemia intended to reflect hyperglycaemic conditions in clinical stroke patients. It is well accepted that hyperglycaemia in stroke patients is of a chronic nature (over extended periods of time) due to lifestyle factors. I question if an acute hyperglycaemic event presented by the Authors is representative of the additional physiological parameters assosciated with chronic hyperglycaemia in stroke patients. At a minimum, this should be discussed in the discussion.

We thank you for this comment. In this study, we chose to test HDL therapy for stroke in acute HG conditions. HG in stroke patients is not only due to diabetes and chronic exposure to high glucose levels. Acute per-ischemic HG, regardless of pre-existing diabetic status, is an important factor for poor outcome after stroke [5]. This state of HG during stroke is likely due to stress and to a non-fasting state [6]. In this work, we sought to test a single parameter (HG) independently of its cause. Chronic HG may cause other changes such as glycation that may produce specific effects. We therefore decided to test only acute HG, without other comorbidities. This condition represents a significant proportion of patients with acute ischemic stroke. The use of chronic models of hyperglycemia due to overweight (such as db/db obese mice or mice fed a high-fat diet) is interesting but in this case, it would be difficult to determine on which parameter HDL might be effective.

To clarify our manuscript, we have added this sentence to the revised discussion section: "This acute elevation of blood glucose is related to a response to stress and a possible non-fasting state [6]."

  1. When reporting the statistics of haemorrhgaic transformation, how can a score of >4 be achieved if the HT score only goes to 4?

We agree with reviewer #2 that this section is inaccurate. We have added this sentence to the revised version: "This score was obtained by adding the scores from 0 to 4 on the 5 brain slices analyzed, which gives a score ranging from 0 to 20".

  1. When determining if HDL is neuroprotective, it would be advisable that a dose reponse be performed, especially when examining HDL's efficacy in a new model (hyperglycaemia + MCAO). Using only a single dose and stating HDL is not neuroprotective undervalues the potential efficacy HDLs could have if a higher dose was used. 

We agree that a dose-response would have been appropriate to state that HDL therapy was not protective under our experimental conditions. We chose this concentration of HDL to be consistent with injectable amounts in humans. Recent studies using CER-001 reported that concentrations as low as 2mg/kg were sufficient to mobilize unesterified cholesterol into the HDL fraction [7]. The same company tested HDL infusion at 8 mg/kg in familial hypercholesterolemic patients [8]. Finally, in our first study in rats, we showed that human plasma HDLs at 10 mg/kg were sufficient to limit infarct volume [2]. On the basis of these data, we tested this concentration in mice. The lack of beneficial effects in the current study underscores the aggravating effect of hyperglycemia and provides us with a good model to improve HDL functionality via their enrichment with additional protective molecules (eg, antioxidants).

"Finally, we chose to test a single dose of HDL as therapeutic. We established this dose based on our previous work and studies conducted in the clinical setting."

  1. Why was an intra-carotid injection used to administer HDLs over IV or IP? I have reservations that the bolus injection (versus slow infusion) of either HDLs or saline could contribute to haemorrhagic transformation post-reperfusion due to significant pressure differentials in a compromised vascular enviroment. This becomes an important point when demonstrating HT values are similar between treatment and vehicle. 

Thank you for this comment. We chose to administer HDL directly into the artery in order to study an interesting route of administration in clinical practice. Indeed, many drugs are injected directly into the carotid artery (eg, rt-PA for stroke or nimodipine for SAH). We cannot exclude a direct effect of injection on bleeding complications. However, this potential bias is mitigated by the fact that both groups received intra-arterial injection. Moreover, our results are consistent with those obtained in our previous work on bleeding complications in acute HG [9]. The scores of hemorrhagic transformations were similar while there was no intra-arterial injection in that work. We mentioned this caveat in the discussion section as a limitation.

  1. I do not understand the value of examining IgG independent of infarct when there is an implied dependency of loss of BBB integrity within the region of the infarct by nature of cell death; this is supported by similar % area values for infarct and IgG presented in Fig 2). Ideally, IgG should be examined and correlated with infarct within the same animal. To perform this, I would recommend infarct evaluation via MRI followed by IgG staining. I understand this is outside the scope of the current study, however, the major comment is required to be addressed. 

We totally agree with the Reviewer #3. This is a limitation of our study. The methods used to quantify IgG infiltration (immunofluorescence staining) and infarct volume with TTC staining are not compatible. We have addressed this comment in the limitation part of our discussion section.

  1. Paigen, B.; Holmes, P.A.; Mitchell, D.; Albee, D. Comparison of atherosclerotic lesions and HDL-lipid levels in male, female, and testosterone-treated female mice from strains C57BL/6, BALB/c, and C3H. Atherosclerosis1987, 64, 215-221, doi:10.1016/0021-9150(87)90249-8.
  2. Lapergue, B.; Moreno, J.A.; Dang, B.Q.; Coutard, M.; Delbosc, S.; Raphaeli, G.; Auge, N.; Klein, I.; Mazighi, M.; Michel, J.B., et al. Protective effect of high-density lipoprotein-based therapy in a model of embolic stroke. Stroke; a journal of cerebral circulation 2010, 41, 1536-1542, doi:10.1161/STROKEAHA.110.581512.
  3. Tran-Dinh, A.; Levoye, A.; Couret, D.; Galle-Treger, L.; Moreau, M.; Delbosc, S.; Hoteit, C.; Montravers, P.; Amarenco, P.; Huby, T., et al. High-Density Lipoprotein Therapy in Stroke: Evaluation of Endothelial SR-BI-Dependent Neuroprotective Effects. Int J Mol Sci 2020, 22, doi:10.3390/ijms22010106.
  4. Gordon, S.M.; Li, H.; Zhu, X.; Shah, A.S.; Lu, L.J.; Davidson, W.S. A comparison of the mouse and human lipoproteome: suitability of the mouse model for studies of human lipoproteins. J Proteome Res 2015, 14, 2686-2695, doi:10.1021/acs.jproteome.5b00213.
  5. Kim, J.T.; Jahan, R.; Saver, J.L.; Investigators, S. Impact of Glucose on Outcomes in Patients Treated With Mechanical Thrombectomy: A Post Hoc Analysis of the Solitaire Flow Restoration With the Intention for Thrombectomy Study. Stroke 2016, 47, 120-127, doi:10.1161/STROKEAHA.115.010753.
  6. Capes, S.E.; Hunt, D.; Malmberg, K.; Pathak, P.; Gerstein, H.C. Stress hyperglycemia and prognosis of stroke in nondiabetic and diabetic patients: a systematic overview. Stroke; a journal of cerebral circulation 2001, 32, 2426-2432.
  7. Keyserling, C.H.; Barbaras, R.; Benghozi, R.; Dasseux, J.L. Development of CER-001: Preclinical Dose Selection Through to Phase I Clinical Findings. Clin Drug Investig 2017, 37, 483-491, doi:10.1007/s40261-017-0506-3.
  8. Kootte, R.S.; Smits, L.P.; van der Valk, F.M.; Dasseux, J.L.; Keyserling, C.H.; Barbaras, R.; Paolini, J.F.; Santos, R.D.; van Dijk, T.H.; Dallinga-van Thie, G.M., et al. Effect of open-label infusion of an apoA-I-containing particle (CER-001) on RCT and artery wall thickness in patients with FHA. J Lipid Res 2015, 56, 703-712, doi:10.1194/jlr.M055665.
  9. Couret, D.; Bourane, S.; Catan, A.; Nativel, B.; Planesse, C.; Dorsemans, A.C.; Ait-Arsa, I.; Cournot, M.; Rondeau, P.; Patche, J., et al. A hemorrhagic transformation model of mechanical stroke therapy with acute hyperglycemia in mice. J Comp Neurol 2018, 526, 1006-1016, doi:10.1002/cne.24386.

Reviewer 3 Report

In the manuscript of Couret et al. the authors test HDL therapy in a mouse model of cerebral ischemia accompanied by acute hyperglycemia, on the outcomes of cerebral infarct size, hemorrhagic transformation, and inflammation. The results do not show specific benefits of HDL treatment.

General:

Overall, the paper is well designed and written. I very much welcome the publication of results showing that there is no particular benefit from specific treatment. In the current case, this is particularly informative, due to the prior publication of this group, where they show a positive effect of HDL therapy on infarct volume and hemorrhagic complications in normoglycemia.

Unfortunately, this positive control was not included in the design of the current study, and so we can only speculate whether the lack of positive effects after HDL therapy is actually due to hyperglycemia or due to some random effect, such as for example insufficient potency of isolated HDL. This could be clearly demonstrated by appropriate control - normoglycemia. This seems to be a major drawback of this study.

Statistics: the authors have to explain the following discrepancy: in the methods (lines 278-282) the authors explain the calculation of the minimum required number of animals in each group (8 per group) in order to statistically detect a relative difference of >25% in the HT score. Although the number of animals examined was greater than the minimum (9 and 11) and also the difference in HT score was greater than 25% (7.5 to 10), the authors nevertheless did not detect a statistically significant difference (p = 0.28) (lines 105- 107). What data dispersion (SD) was taken into account in the calculation?

Although the discussion is well set, the reader should make an effort to understand the whole concept, and should look for information in the cited literature. However, the last sentence in the last para, which is usually meant as a conclusion, is somewhat disturbing:  “The use of HDL enrich with antioxidants could be a therapeutical approach to enhance their neuroprotective potential.”,  as this is not supported by the results of this article - so it should be omitted from the last paragraph.

Minor Specific comments:

There are some 'bugs' in all the figures that need to be corrected: (eg., red box on Fig. 1 – missing?; Figs. 1-4: ?-marks; legend to Fig. 5B: »Bar = 80 μm for low magnification pictures« - but there is only one image.)

The cell count on the graphs (Figures 4C and 5A) is not informative enough - it must also be expressed in some relative value, e.g. as a proportion of all cells examined.

Author Response

Reviewer 3

In the manuscript of Couret et al. the authors test HDL therapy in a mouse model of cerebral ischemia accompanied by acute hyperglycemia, on the outcomes of cerebral infarct size, hemorrhagic transformation, and inflammation. The results do not show specific benefits of HDL treatment.

General:

Overall, the paper is well designed and written. I very much welcome the publication of results showing that there is no particular benefit from specific treatment. In the current case, this is particularly informative, due to the prior publication of this group, where they show a positive effect of HDL therapy on infarct volume and hemorrhagic complications in normoglycemia.

Unfortunately, this positive control was not included in the design of the current study, and so we can only speculate whether the lack of positive effects after HDL therapy is actually due to hyperglycemia or due to some random effect, such as for example insufficient potency of isolated HDL. This could be clearly demonstrated by appropriate control - normoglycemia. This seems to be a major drawback of this study.

We agree with the Reviewer #3. In order to take this notion into account as a limitation of our study, we have added this sentence in the discussion section: “First, we chose to test HDL only in the acute hyperglycemic situation. Indeed, we had already demonstrated their efficacy in previous works in a standard glycemic situation. This was done to reduce the use of animals with an additional control group and to comply with the 3R rule.”

Statistics: the authors have to explain the following discrepancy: in the methods (lines 278-282) the authors explain the calculation of the minimum required number of animals in each group (8 per group) in order to statistically detect a relative difference of >25% in the HT score. Although the number of animals examined was greater than the minimum (9 and 11) and also the difference in HT score was greater than 25% (7.5 to 10), the authors nevertheless did not detect a statistically significant difference (p = 0.28) (lines 105- 107). What data dispersion (SD) was taken into account in the calculation?

We agree with the Reviewer #3 that these sentences are confusing.  The HT score ranged from 0 to 20 (adding five brain slices 0-4). We also took into account the mortality rate of these models (30%). The difference we would have needed to have a significant result would have been 5 points out of 20. But here we have an insufficient difference (2.5/20). In order to improve the understanding of the reader, we have added this sentence in the method section: “This score was obtained by adding the scores from 0 to 4 on the 5 brain slices analyzed, which gives a score ranging from 0 to 20”

Although the discussion is well set, the reader should make an effort to understand the whole concept, and should look for information in the cited literature. However, the last sentence in the last para, which is usually meant as a conclusion, is somewhat disturbing:  “The use of HDL enrich with antioxidants could be a therapeutical approach to enhance their neuroprotective potential.”,  as this is not supported by the results of this article - so it should be omitted from the last paragraph.

We totally agree with the Reviewer #3. This sentence has been removed from the revised version of the manuscript.

Minor Specific comments:

There are some 'bugs' in all the figures that need to be corrected: (eg., red box on Fig. 1 – missing?; Figs. 1-4: ?-marks; legend to Fig. 5B: »Bar = 80 μm for low magnification pictures« - but there is only one image.)

Thank you for this note. Corrections have been made.

The cell count on the graphs (Figures 4C and 5A) is not informative enough - it must also be expressed in some relative value, e.g. as a proportion of all cells examined.

For quantification in Figures 4C and 5A, we analyzed the number of positive cells per field. For Figure 4C, we chose 2 different fields, one in the cortex and one in the striatum. For Figure 5A, we chose a single field in the striatum. The same field was analyzed for each mouse.

Round 2

Reviewer 1 Report

I have no additional comments for the authors or the editor. The authors have answered the raised questions reasonably well. 

Final spell check and figure proof check (style, size) to fit the journal is needed. 

Author Response

Thank you for your comments which have improved the quality of our work. 

Reviewer 2 Report

Authors have adequately addressed review comments. 

Author Response

(The authors gave the same response as above.)

Reviewer 3 Report

  1. The answer regarding the lack of positive control is not satisfactory. It is not enough to just acknowledge the limitations. Authors should discuss the consequences of the results obtained without proper control - whether the results are valid, whether other scientists can rely on them, what the possible biases are, and so on. Discuss whether it is possible that the lack of positive effects after HDL therapy is due to a factor other than acute HG.

Argument of the lack of control group due to "reduction of animal use and compliance with rule 3R." does not hold, as the definition of the second R (reduction) is: “Appropriately designed and analyzed animal experiments that are robust and reproducible, and truly add to the knowledge base.”, which is exactly what is missing in this study.

  1. What exactly do the authors mean by the term “... a standard glycemic situation”?? Would that be a normal blood glucose level?

  1. The response to the discrepancy in the statistics is unsatisfactory: I was already well acquainted with all the facts repeated in the authors' response (30% mortality, HT score from 0-20 estimate, ... etc.). However, I learned from the answer that the authors misinterpret “...in order to statistically detect a relative difference of >25% in the HT score.« The relative difference between the two samples means that the two samples will differ by 25% in some value, not that the difference between them will be 25% of the maximum possible value. So, if the authors wanted to detect a difference representing 25% of the maximum value, then the correct description would be "... "... to statistically detect an absolute difference of 5 in the HT score.".

In order for the full description of the statistics to have any weight at all, other data are needed. Thus, the authors still did not provide an answer to the question: What data dispersion (SD) was taken into account in the calculation? (of sample size).

  1. Supplemental Figure S1. The correlation of the measurements of the two operators is indeed very high, but this is not a measure of the repeatability or accuracy of the measurements. It is very clear that both operators reported quite different measurement results. Therefore, it would be much more informative to give the average difference between the measurements of the two operators and to show that the differences are not statistically significant.

Author Response

Reviewer 3
1. The answer regarding the lack of positive control is not satisfactory. It is not enough to just
acknowledge the limitations. Authors should discuss the consequences of the results
obtained without proper control - whether the results are valid, whether other scientists can
rely on them, what the possible biases are, and so on. Discuss whether it is possible that the
lack of positive effects after HDL therapy is due to a factor other than acute HG.
Argument of the lack of control group due to "reduction of animal use and compliance with rule 3R."
does not hold, as the definition of the second R (reduction) is: “Appropriately designed and analyzed
animal experiments that are robust and reproducible, and truly add to the knowledge base.”, which
is exactly what is missing in this study.

We agree with the Reviewer 3. The absence of a group that was not subjected to induced
hyperglycemia is not sufficiently justified in our work. Indeed, this limitation is now discussed to
allow the reader a more informed interpretation of our results. In our previous work by Tran-Dinh et
al., we demonstrated the efficacy of HDL injection on ischemic injury without induction of
hyperglycemia [1]. In this study, different conditions were used to achieve significant cerebral
infarction (4-hour transient intraluminal middle cerebral artery occlusion-MCAO). Under
hyperglycemic conditions, the maximum duration of occlusion was set at 90 min [2,3] to avoid
excessive mortality. However, a 90-min MCAO do not produce a large enough infarct to test the
effect of a potential treatment such as HDL particles. Because this experimental group was not
strictly under the same conditions, we could not use it in the present study as a normoglycemic
control group. This point is now discussed in more detail in the revised version of the manuscript, to
highlight one major limitation of our study: " One of the limitation of this study is the absence of a
normoglycemic group treated with the same batch of HDLs, which would attest to their protective
effects in the absence of induced hyperglycemia. Despite a trend toward reduced infarct volume and
IgG infiltration (Figure 2A,B) in the HDL-treated group, we cannot conclude with certainty whether
this lack of protection is due to the hyperglycemic condition or to a lack of functionality of the HDLs
tested. An additional group without hyperglycemia induction would have been appropriate, but
would have had to be performed in a different experimental setting (4-hour instead of 1.5-hour
MCAO) to produce a cerebral infarct with of sufficient volume to test the effects of HDLs [2,3].
However, reproducibility of HDL isolation was tested in terms of composition and potential
contamination by SDS-PAGE and Coomassie blue staining. In future studies, a normoglycemic control
group and/or in vitro experiments attesting to the functionality of HDLs would be required".

2. What exactly do the authors mean by the term “... a standard glycemic situation”?? Would
that be a normal blood glucose level?

We agree with the Reviewer 3, that the term "standard" is not appropriate. We meant “normal”
blood glucose levels. We have corrected this sentence in the revised version of the manuscript.

3. The response to the discrepancy in the statistics is unsatisfactory: I was already well
acquainted with all the facts repeated in the authors' response (30% mortality, HT score from
0-20 estimate, ... etc.). However, I learned from the answer that the authors misinterpret
“...in order to statistically detect a relative difference of >25% in the HT score.« The relative
difference between the two samples means that the two samples will differ by 25% in some
value, not that the difference between them will be 25% of the maximum possible value. So,
if the authors wanted to detect a difference representing 25% of the maximum value, then
the correct description would be "... "... to statistically detect an absolute difference of 5 in
the HT score.".
In order for the full description of the statistics to have any weight at all, other data are needed.
Thus, the authors still did not provide an answer to the question: What data dispersion (SD) was
taken into account in the calculation? (of sample size).

Thank you for this comment that improves quality of our study. For the calculation of the number of
mice needed, we took an absolute difference of 5 with a known mean score of 10 and an SD of 3.5.
With a power of 80% and a firs-order risk of error of 5%, we need 8 mice per group. The materials
and methods section has been revised.

4. Supplemental Figure S1. The correlation of the measurements of the two operators is indeed
very high, but this is not a measure of the repeatability or accuracy of the measurements. It
is very clear that both operators reported quite different measurement results. Therefore, it
would be much more informative to give the average difference between the measurements
of the two operators and to show that the differences are not statistically significant.
As requested by the Reviewer #3, we now present the average differences between the two
operators (See attachment).
